# Diagnostic Performance of Multi-Detector Computed Tomography Arthrography and 3-Tesla Magnetic Resonance Imaging to Diagnose Experimentally Created Articular Cartilage Lesions in Equine Cadaver Stifles

**DOI:** 10.3390/ani13142304

**Published:** 2023-07-14

**Authors:** Nico M. Bolz, José Suárez Sánchez-Andrade, Paul R. Torgerson, Andrea S. Bischofberger

**Affiliations:** 1Equine Hospital, Vetsuisse Faculty, University of Zurich, 8057 Zurich, Switzerland; 2Clinic for Diagnostic Imaging, Vetsuisse Faculty, University of Zurich, 8057 Zurich, Switzerlandabischofberger@vetclinics.uzh.ch (A.S.B.); 3Section of Veterinary Epidemiology, Vetsuisse Faculty, University of Zurich, 8057 Zurich, Switzerland

**Keywords:** computed tomography, magnetic resonance imaging, cartilage lesion, arthroscopy, stifle joint, horse

## Abstract

**Simple Summary:**

In order to prevent the onset of osteoarthritis, timely detection of cartilage damage is essential. We hypothesized that the diagnostic performance of computed tomography arthrography (CTA) would be superior to that of 3-Tesla MRI when diagnosing cartilage lesions in the stifle joint of horses. A total of 79 cartilage defects were created arthroscopically in 15 cadaver stifles from adult horses, with 68 sites serving as negative controls. A radiologist unaware of the lesion distribution evaluated the images, followed by macroscopic evaluation including measuring lesion surface area, depth, and volume. The sensitivity and specificity of MRI and CTA were calculated and compared between modalities. The sensitivity values of CTA (53%) and MRI (66%) were not significantly different (*p* = 0.09). However, the specificity of CTA (66%) was significantly greater compared to MRI (52%) (*p* = 0.04). In conclusion, CTA achieved a similar diagnostic performance compared to high-field MRI in detecting small experimental cartilage lesions. Despite this, CTA showed a higher specificity than MRI; thus, CTA was more accurate in diagnosing normal cartilage. Small lesion size was a discriminating factor for lesion detection. In a clinical setting, CTA may be preferred over MRI due to higher availability and easier image acquisition.

**Abstract:**

Background: The purpose of the study was to determine the diagnostic performance of computed tomographic arthrography (CTA) and 3-Tesla magnetic resonance imaging (MRI) for detecting artificial cartilage lesions in equine femorotibial and femoropatellar joints. Methods: A total of 79 cartilage defects were created arthroscopically in 15 cadaver stifles from adult horses in eight different locations. In addition, 68 sites served as negative controls. MRI and CTA (80–160 mL iodinated contrast media at 87.5 mg/mL per joint) studies were obtained and evaluated by a radiologist unaware of the lesion distribution. The stifles were macroscopically evaluated, and lesion surface area, depth, and volume were determined. The sensitivity and specificity of MRI and CTA were calculated and compared between modalities. Results: The sensitivity values of CTA (53%) and MRI (66%) were not significantly different (*p* = 0.09). However, the specificity of CTA (66%) was significantly greater compared to MRI (52%) (*p* = 0.04). The mean lesion surface area was 11 mm^2^ (range: 2–54 mm^2^). Greater lesion surface area resulted in greater odds of lesion detection with CTA but not with MRI. Conclusions: CTA achieved a similar diagnostic performance compared to high-field MRI in detecting small experimental cartilage lesions. Despite this, CTA showed a higher specificity than MRI, thus making CTA more accurate in diagnosing normal cartilage. Small lesion size was a discriminating factor for lesion detection. In a clinical setting, CTA may be preferred over MRI due to higher availability and easier image acquisition.

## 1. Introduction

The timely detection of cartilage damage in the joint disease process is essential to allow prompt treatment, restore joint functionality, and prevent or delay the onset of osteoarthritis (OA) [1,2]. It has been reported that if there are radiographic signs of OA pre-operatively, this indicates a poor prognosis for the restoration of joint functionality, even after arthroscopic cartilage debridement [3].

Cartilage lesions in the equine stifle diagnosed with arthroscopy on the medial femoral condyles have been suspected to be the sole cause of stifle pain in horses, but they are more commonly found in combination with other lesions such as meniscal lesions, subchondral bone cysts, and desmitis of the cruciate, collateral, and meniscotibial ligaments [4,5]. 

The severity of these articular cartilage lesions may range from superficial fibrillation or chondromalacia to partial- or full-thickness defects and the formation of cartilage flaps [3,4,5,6]. In younger horses, osteochondrosis dissecans is another common source of articular cartilage defects, most commonly affecting the femoropatellar joint (FPJ) [5]. 

The classical diagnostic work-up of equine stifle disease using intra-articular anesthesia, radiography, and ultrasonography has reported limitations in the early detection of cartilage defects [3,4,7,8,9,10]. While arthroscopy is considered the reference to evaluate cartilage damage in the stifle, it is invasive, and a complete evaluation of the articular surface is impossible due to anatomical constraints [5,11,12,13]. As a result, in horses, three-dimensional imaging has gained popularity in recent years. Technical improvements, fast scanning times, and the increased availability of computed tomography (CT) scanners with large bore diameters have made CT a more sought-after diagnostic tool, especially with the possibility to combine it with arthrography (CTA) and subsequent stifle arthroscopy [14,15]. 

The CT anatomy of the equine stifle has been described [16]. The ideal intra-articular contrast concentration to detect articular cartilage defects in equine carpal joints using CT has been reported; however, no comparable reports exist for the equine stifle [17]. Protocols for contrast medium volume and concentration in the stifle vary widely: Reported volumes for each of the femorotibial joint compartments (FTJ) range from 60–80 mL, and for the FPJ from 80–150 mL at varying iodine concentrations (35–150 mg iodine/mL). In some stifles, total volumes of up to 360 mL were needed for maximal joint distention [16,18,19]. 

While studies have shown good results in detecting cartilage defects using CTA in equine carpi and fetlocks, in a case series describing 16 horses with pain localized to the stifle, only a single cartilage lesion was detected with CTA [15,20,21]. Similarly, a cadaveric study investigating experimentally induced soft tissue lesions in equine stifles showed comparable sensitivity and specificity values of high-field MRI and CTA; however, no cartilage lesions were included in the study [19].

Magnetic resonance imaging (MRI) has better soft tissue contrast than CT and is the modality of choice to evaluate human knee joint disease. In humans, the main reason for its preferability is likely the lack of exposure to ionizing radiation and the need for arthrocentesis, rather than a better diagnostic performance [22]. The sensitivity of MRI ranges from 15–96% with a specificity above 90%, both being dependent on the scanning protocol and the magnet strength used. [23,24]. 

In horses, CT has several advantages over MRI including fast acquisition times resulting in shorter general anesthesia times and lower costs, better spatial resolution, improved multiplanar reconstruction, and a bigger gantry size [15,18]. The normal MRI anatomy of the equine stifle [25], as well as two case series, have been published: Santos et al. reported the detection of cartilage lesions in the stifle using low-field MRI in seven asymptomatic horses and concluded that low-field MRI is suitable to detect cartilage lesions [26]. On the other hand, Waselau et al. reported low-field MRI findings in clinical cases including bone marrow lesions, osseous cyst-like lesions, cruciate ligament desmopathy, and meniscal tearing, but no cartilage lesions were detected [27]. While CTA has been shown to be superior in detecting cartilage lesions in the equine fetlock and carpus compared to MRI [20,21], no comparable studies exist for the stifle. 

The goal of this study was to determine the sensitivity and specificity of helical multi-detector CTA and high-field MRI to detect experimentally created articular cartilage lesions in equine stifle joint cadavers. We hypothesized that the diagnostic performance of CTA would be superior to 3-Tesla MRI when diagnosing cartilage lesions in eight clinically relevant locations. 

## 2. Materials and Methods

Hind limbs of adult warmblood and Thoroughbred horses without hind limb lameness, aged 5–22 years, euthanized due to reasons unrelated to the study, were collected at the Vetsuisse Faculty, University of Zurich, following the obtainment of owner consent. The limbs were disarticulated at the coxofemoral joint within 24 h of euthanasia, labeled with a case-specific number, and frozen in a 90° flexed position at −28°. 

### 2.1. Arthroscopy Protocol

The frozen limbs were defrosted for a minimum of 24 h in tap water to reach room temperature and positioned with the femoral head fixed on a table and the hoof fixed to a hoist mounted on the ceiling. Flexion and extension of the limb could be adjusted by lowering or raising the hoist during arthroscopy. All arthroscopies were performed by or under the supervision of a board-certified surgeon (AB) using a rigid arthroscope with a diameter of 4 mm and a 30° forward-angled lens system. The arthroscopic access to the joint compartments was as follows: In 90° flexion, an 8 mm skin incision with a No. 11 blade was made through the skin and continued through the fascia into the middle patellar ligament 2 cm proximal to the tuberositas tibia. Penetration of the medial FTJ capsule was achieved by inserting the arthroscopic sleeve containing the conical obturator in a slight caudoproximoaxial direction. To access the lateral FTJ, the conical obturator was reintroduced in the arthroscopic sleeve, the sleeve pulled back as far as not to leave the incision into the middle patellar ligament and then redirected in a caudoproximoaxial direction, aiming more towards the lateral direction. To enter the caudal part the stifle, joints were extended to 120° flexion, and via the cranial approach the scope was pushed as far abaxially as possible to improve caudal joint distention. The medial caudal arthroscopic portal was made about 6–8 cm caudal to the medial collateral ligament and 1 cm proximal to the level of a line between the palpable tibial plateau and tibial condyle, cranial to the saphenous vein [28]. Access to the lateral caudal FTJ was made in the same fashion, but to avoid the peroneal nerve and to be proximal to the popliteal tendon, the portal was placed 2.5 cm proximal to the tibial plateau and 3 cm caudal to the lateral collateral ligament. The FPJ was accessed via a portal localized midway between the tibial crest and the patellar apex and between the middle and lateral patellar ligament. After an 8 mm skin incision, the blunt obturator and cannula were directed 45° proximad and introduced into the FPJ. Instrument portals were created under visual control [5].

A thorough examination of all joint compartments to detect pre-existing cartilage lesions was undertaken, with joint distention achieved using normal tap water pumped through a volume- and pressure-controlled arthroscopic pump (Karl Storz Tuttlingen Germany). If a joint showed pre-existing lesions, the joint was excluded from the study. 

Cartilage lesions were made under visual control with size 0 and 00 arthroscopic curettes (Karl Storz Tuttlingen Germany) at predetermined locations. Defect size was randomly determined by the surgeon and was measured macroscopically at a later time point (see below). Lesion distribution was predetermined by a randomized list as follows: cranial aspect of the medial and lateral femoral condyle: 0–2 lesions each; caudal aspect of the medial and lateral femoral condyle: 0–1 lesion each; lateral and medial femoral trochlear ridge and intertrochlear femoral groove: 0–1 lesion each; and facies articularis patellae: 0–1 lesion. Sites that were left without lesions served as negative controls. Lesions were created on the cranial aspect of the femoral condyle, best visualized with a cranial arthroscopic approach over the cranial pouch of the medial or lateral FTJ close to the meniscus. The caudal femoral condyle lesions were located on the caudal femoral condyle and best accessed via a caudomedial or lateral approach, placing them on the abaxial to caudal face of the condyle (Figure 1). 

The joint capsule and the skin were closed separately in a tight, simple, continuous pattern using USP 2-0 absorbable sutures. Following lesion creation, the limbs were refrigerated at 7° and underwent MRI and CT at room temperature within 24 h. 

### 2.2. Diagnostic Imaging

MRI was performed in a 3-Tesla MRI scanner with a dStream Torso coil (Philips Ingenia, Philips AG, Zurich, Switzerland). The MRI protocol (sequences, planes, and acquisition parameters) is shown in Appendix A. The limbs were positioned in lateral recumbency. After the MRI examination, the stifle joints were prepared for CTA.

First, all three stifle joint compartments were injected with iodinated contrast medium (Optiray 350, 350 mg iodine/mL, Mallinckrodt AG, Steinhausen, Switzerland) and diluted 1:3 with 0.9% NaCl to a final concentration of 87.5 mg/mL. Needle placement for the FPJ was midway between the tip of the patella and the tibial crest between the middle and lateral patellar ligament, aiming proximad under the patella. For the medial FTJ the needle was inserted between the medial patellar ligament and the medial collateral ligament about 2 cm proximal to the tibial plateau, and for the lateral FTJ the needle was inserted caudal to the lateral patellar ligament and also about 2 cm proximal to the tibial edge [29]. The joint compartments were injected with 80 mL in each FTJ and 160 mL in the FPJ, followed by manual flexion of the limbs over 2 min. After no more than 20 min, the CT images were acquired using a multidetector 16-slice scanner (Philips 16 Brilliance, Philips AG, Zurich, Switzerland) in helical mode acquisition, with a pitch of 0.55, 140 kV, 350 mAs, a field of view of 176 mm, and a matrix of 300 × 300. The slice thickness was 0.6 mm in the bone and soft tissue algorithm. In the CT scanner, the limbs were not placed uniformly the same way, resulting in some limbs positioned in lateral and others in medial recumbency. Following the CT scan, the limbs were directly frozen until the day of the macroscopic evaluation 7–40 days following the imaging procedure.

### 2.3. Gross Analysis

The defrosted limbs were carefully disarticulated in the stifle joint by one of the authors (N.B.). The depth and width of cartilage lesions were measured grossly using a caliper (Tesa Twin Cal, TESA, Renens, Switzerland, error margin 0.03 mm). All articular surfaces were photographed. 

### 2.4. Image Analysis

A board-certified radiologist (JSS) reviewed all images using a diagnostic workstation and medical imaging software (Intellispace PACS Radiology 4.4553.0, Phillips Healthcare, Zurich, Switzerland) twice at two different time points 6 months apart. During the second image analysis, the radiologist had access to the first round of assessments, and corrected his assessment if deemed necessary. The time point of 6 months was chosen randomly according to the work schedule and availability of the radiologist. The final results after the second analysis were used for further calculations. The observer was unaware of the presence, location, and thickness of the cartilage lesions at both time points, and they evaluated the stifles in all planes using multiplanar reconstruction. Lesions were defined as a local disruption of cartilage integrity, visible as a focal contrast column in otherwise unaffected cartilage. The exact lesion location was noted in a Microsoft Excel sheet. 

### 2.5. Data Analysis

Data were stored in a Microsoft Excel sheet. The sensitivity and specificity values of CTA and MRI were calculated for all lesions independent of the joint and the lesion groups (medial and lateral FTJ and FPJ). The median, mean ± standard deviation lesion depth, lesion surface area, and lesion volume were calculated for all lesions from the macroscopic measurements. The calculation of the approximated lesion surface area and volume was performed with the simplification of the lesion into a rectangular shape and to have a cuboidal volume, neglecting the true round or dome shape. Then, the mean thickness of correctly identified lesions in the CTA and MRI scans was compared to lesions not detected correctly.

Arthroscopy was the reference for whether a lesion was created or not. Using McNemar’s test, the sensitivity and the specificity values between the modalities were compared. Binary logistic regression was used to evaluate the association between the independent variables (joint (FTJ or FPJ)) and the covariates (lesion depth, lesion surface area, and lesion volume) on CTA or MRI diagnosis (no lesion or lesion present). The mean lesion surface areas of detected lesions and non-detected lesions were compared using *t*-tests. The optimal Youden index point was calculated for MRI and CTA. Statistical analyses were performed using R (R Foundation for Statistical Computing, Vienna, Austria. URL https://www.R-project.org/ using the MASS, car, and lme4 packages, accessed on 12 December 2023). A *p*-value < 0.05 was considered significant.

## 3. Results

Arthroscopy of all three joint compartments was possible, and the visible cartilage surfaces were without abnormal findings in 19 cadaver horse stifles harvested from 11 warmblood horses (6 mares and 5 geldings). The mean age was 14.2 ± 4.85 years (range: 5–22 years). 

One limb was excluded due to poor quality of the MRI study related to poor positioning, while the rest of the MRI studies were considered good to excellent by the radiologist. In two stifles, non-uniform contrast medium distribution was detected in the CT studies. In one limb only the lateral FTJ was affected. In this limb only the lateral FTJ was removed from the calculations, and the medial FTJ and the FPJ could be used. In the other limb, all three joints had to be excluded. The radiologist could not assess two limbs due to problems with image storage, and these limbs had to be removed from the study. Finally, 14 complete stifles and one stifle with the lateral FTJ excluded could be used for the calculations. 

Intra-articular air accumulation was detected on both CTA and MRI images and could not be avoided because lesions needed to be created before the scans. 

Overall, 79 lesions were created. Examples of the cartilage lesions made are depicted in Figure 2. Examples of CTA and MRI images of cartilage lesions are shown in Figure 3. In 68 sites no lesion was created, and they served as negative controls. In the medial FTJ, 27 lesions were created, and 18 sites were left without a lesion and served as negative controls. In the lateral FTJ, 17 lesions were created, and 25 sites were left as negative controls. In the FPJ, 35 lesions were created, and 25 sites were left as negative controls. 

The overall sensitivities of CTA (53.2%) and MRI (65.8%) were not significantly different (*p* = 0.087). However, the overall specificity was significantly higher for CTA (66.2%) compared to MRI (51.5%) (*p* = 0.04). The joint-specific sensitivity and specificity values are shown in Table 1 and were not significantly different between CTA and MRI. In both modalities, the sensitivity was highest in the FPJ, where the specificity was 56% for CTA and 36% for MRI. In the FTJ the sensitivity values for CTA and MRI were 48% medially and 35% laterally; and 63% medially and 47% laterally, respectively. The sensitivity was higher medially than laterally in both modalities. In contrast, the specificity values of CTA and MRI were very similar for each FTJ. 

Macroscopic measurements were obtained from 59 sites; 20 lesions could not be measured. The main reason for inability to measure a lesion was that cartilage damage occurred during disarticulation to a degree that the lesions could not be measured reliably. Overall, the mean macroscopically measured lesion depth was 2.1 mm (range: 0.8–5.4 mm, median 2.0 mm), the mean lesion surface area was 11 mm^2^ (range: 2–54 mm^2^, median 8 mm^2^), and the mean lesion volume was 24 mm^3^ (range: 5–126 mm^3^, median 20 mm^3^). In the logistic regression model, only lesion surface area (*p* = 0.03), not joint, lesion depth, or lesion volume, had a significant effect on lesion detection by CTA. In MRI neither joint, lesion surface area, volume, nor depth had a significant effect on whether a lesion could be detected or not. The mean surface area of lesions detected by CTA was significantly greater (13.8 mm^2^) compared to those not detected (8.1 mm^2^) (*p* = 0.014) (Figure 4). In MRI there was no significant difference in the mean surface areas of detected (12.1 mm^2^) and non-detected lesions (9.6 mm^2^) (*p* = 0.301) (Figure 4). 

Cut-off values were calculated for lesion surface area and optimal sensitivity, and specificity values were determined, representing the Youden index point on an empirical ROC curve (Figure 5). In CTA, with the cut-off for lesion surface area at 3.9 mm^2^, a sensitivity of 61% and a specificity of 70% could be reached. For MRI, a cut-off value of 4.53 mm^2^ for lesion surface area resulted in a sensitivity of 55% and a specificity of 80%. 

## 4. Discussion

The goal of this study was to compare the diagnostic performance of CTA to a high-field MRI method in detecting arthroscopically created small articular cartilage lesions within the stifle joints of equine cadavers. We hypothesized that these lesions would be more reliably detected using the CTA than MRI, as has been shown for other equine joints [20,21]. This hypothesis was only partially proven: no significant difference was found between the sensitivities of the modalities; however, CTA showed a small but statistically significant higher specificity than MRI in detecting artificially created cartilage lesions. 

While the sensitivity of CTA (53%) in this study was lower than the reported values for the equine carpal joints (70%) and equine fetlock joint (82%), MRI performed with a sensitivity of 66%, which was better than in the carpus (33%) and also than in the fetlock (41%) [20,21]. Despite not being significant, the difference in sensitivity between the two modalities in our study, with a *p*-value of 0.09, is a finding close to significance and might have become significant with a larger sample size.

The image quality of CTA is affected by the effectiveness of contrast agent application. For the lesions to be visible in CTA images, sufficient contrast agent at an adequate concentration has to accumulate in the cartilage defect. Reliable injection of contrast agent into the carpal joints and its homogenous distribution have been shown, reaching optimal detail with a small volume of 10 mL [17]. The complex equine stifle joint is surrounded by a considerable amount of soft tissue, making reliable injection of all three compartments difficult, and much higher volumes are needed [14,15,30,31]. In an equine clinical study using stifle CTA, in one-third of the cases, an incomplete contrast distribution was reported [14]. Even after correct intra-articular injection, the study design and anatomy of the stifle joint could explain the lower sensitivity in this study for CTA compared to values reported for other equine joints.

Due to the necessity to create the lesions before the imaging procedure, leakage of contrast agent from the arthroscopic portals potentially led to a decrease in contrast agent volume in the joint and decreased the filling pressure. Furthermore, lesions created on the femur condyle could be covered by the menisci when extending the joint for MRI and CTA from its flexed position during arthroscopy. This could then prevent the inflow of contrast agent into these lesions. Further studies would be needed to evaluate whether scanning at different degrees of flexion would improve lesion detection. The contrast agent, having a greater density compared to the normal synovial fluid, was observed to accumulate in the recumbency-dependent lowest-lying joint compartments [18]. The large difference between the CTA sensitivity (35%) and MRI sensitivity (47%) in the lateral FTJ is possibly explained by the influence of gravity, if the lateral compartment was situated uppermost. Additionally, the fact that the lateral FTJ is often more difficult to inject may play a role [30]. 

The use of 3D MRI sequences might explain the comparable sensitivities of CTA and MRI in our study. Three-dimensional sequences such as 3D_T2W_HR, 3D_PDW_SPAIR, and T1W_VSTA_SPAIR have slice thicknesses between 0.55 and 0.7 mm and therefore allow the evaluation of cartilage surfaces with a comparable spatial resolution to CTA. In clinical patients, secondary changes such as signal intensity in the subchondral bone, amongst others, would improve the detection rate of cartilage lesions in MRI and have a lesser impact on CTA [32]. Whether freezing of the cadaver limbs, gas artifacts after arthroscopy and injection of the contrast agent, and ferromagnetic dust left by the surgical instruments after arthroscopic surgery have an influence on the diagnostic accuracy with changes in the signal-to-noise ratio or image distortion of MRI studies is unclear [33,34]. In our study, intra-articular air accumulation made the evaluation of the cartilage-to-synovium interface more challenging in both the CTA and the MRI evaluation. A retrospective unblinded evaluation of the cases to assess the impact of intra-articular gas accumulation was not performed.

Most importantly, in the literature the influence of lesion grade severity on the detectability of cartilage defects has been shown [35]. The different grading systems of cartilage lesions and different diagnostic imaging settings used in different studies result in a high variance in diagnostic performance [36]. This is reflected in the wide range of reported sensitivity (CTA 46–95% and MRI 9–83%) and specificity values (CTA 72–96% and MRI 27–99%) for cartilage lesion detection in human and equine studies [20,21,37,38]. Comparing arthroscopy to 3D MRI in human clinical cases, MRI rules out cartilage damage reliably with a specificity of 95–99% with or without contrast agent injection [23,24,35,39]. This is in contrast to our study, where 3D MRI only reached a specificity of 51.5% when including all lesions. With an optimal cut-off for lesion surface area of 4.53 mm^2^, a specificity as high as 80% and a sensitivity of 55% were reached. Using a cartilage-specific scanning protocol in humans, the sensitivity decreased from 83% in large lesions to as low as 9% for grade 1 lesions on the modified Outerbridge classification [35,40]. 

When removing the smallest lesions from our calculations by applying the optimal lesion surface area of 3.9 mm^2^, the optimal Youden index point to detect lesions had a sensitivity of 61% and a specificity of 70% for CTA (Figure 5). A similar observation was made by Nelson et al., comparing the grading of cartilage lesions during arthroscopy with findings from equine stifle CTA using a standardized semiquantitative grading scale. They report that while six-eighths of the severe cartilage defects were identified, only one-eighth of moderate cartilage defects were detected, and of the four lesions arthroscopically graded as mild, none were detected with CTA [14]. Therefore in both modalities, the sensitivity and specificity increased to a level similar to what has been reported in carpal and fetlock joints for CTA and outperformed the reported MRI results in equine studies [20,21], with results closer to the reported levels in human studies. 

From a clinical aspect, removing grade 1 lesions can be justified, as lesions on the medial femoral condyle need to have a surface area as large as 64 mm^2^ to reliably induce osteoarthritis in horses [41]. On the other hand, in a more recent study lesions as small as 3.1 mm^2^ led to defects that did not heal spontaneously [42]. Therefore, the lesions in the present study, while being on the smaller side with an average surface area of 11 mm^2^, potentially mimic lesions that could cause long-term disease if not detected and treated accordingly.

Limitations of this study include the abovementioned rather small lesions made. The experimental nature of the study resulted, along with the inability to distend the joint cavities properly with the contrast agent due to leakage from the arthroscopic portals, also in intra-articular gas accumulation impeding the evaluation of the cartilage contours. Additionally, the low number of cadaver joints decreased the power of the study. An attempt was made to perform standardized lesion-following a randomized protocol. However, due to the difficulties in performing arthroscopy in the stifle cadavers, we could not follow a protocol that would allow a more distinctive placement and protocolization of the lesions concerning the exact anatomical location—for example, at the middle, distal, or proximal femoral condyle. The degrees of joint flexion, joint distention, and extravasation were slightly variable, influencing the arthroscopic procedure and lesion placement. This may have influenced the lesion detection rate of MRI and CTA and represents a limitation. We were not able to measure all arthroscopically created lesions macroscopically. This could have had an impact on the calculated lesion depth and surface area differences between detected and non-detected lesions. Although they were very experienced, a further limitation is that only one diplomate radiologist was available to evaluate the CT and MRI images. Therefore, no interobserver analysis could be performed. It may be that a consensus of several radiologists would have affected the results.

Whether the 3D MRI sequences used in this study will be introduced into equine practice will mainly depend on the availability of these scanners, the ease of horse placement in the gantry, the time for image acquisition, and the susceptibility to technical artifacts. CTA most likely will have a higher sensitivity and specificity in a clinical setting because joint distention is more easily achieved. Arthroscopy is still seen as the method of choice for a thorough evaluation of cartilage damage in clinical cases [14,23,24,37,39]. Especially, generalized morphologic changes termed chondromalacia (grade 1 on the modified Outerbridge classification) are often detected during arthroscopy in horses. One recent investigation reported decreased long-term outcomes in cases affected by these generalized cartilage changes [6]. Without clear surface defects that could be filled with contrast agent, detection with classical CTA ranges from very limited to impossible. In the future, with dual-energy CT, cationic contrast agents, and quantitative MRI, the diagnostic capability of three-dimensional imaging could improve fast, resulting in a non-invasive method to evaluate even functional properties of equine cartilage [43,44,45,46,47].

## 5. Conclusions

CTA achieved a similar diagnostic performance compared to high-field MRI in detecting small experimental cartilage defects, with sensitivity values of 53% for CTA and 66% for MRI. Despite this, CTA showed a higher specificity (66%) than MRI (52%); thus, CTA was more accurate in diagnosing normal cartilage. Small lesion size was a discriminating factor for lesion detection. In a clinical setting, CTA may be preferred over MRI due to higher availability and easier image acquisition.

## Figures and Tables

**Figure 1 animals-13-02304-f001:**
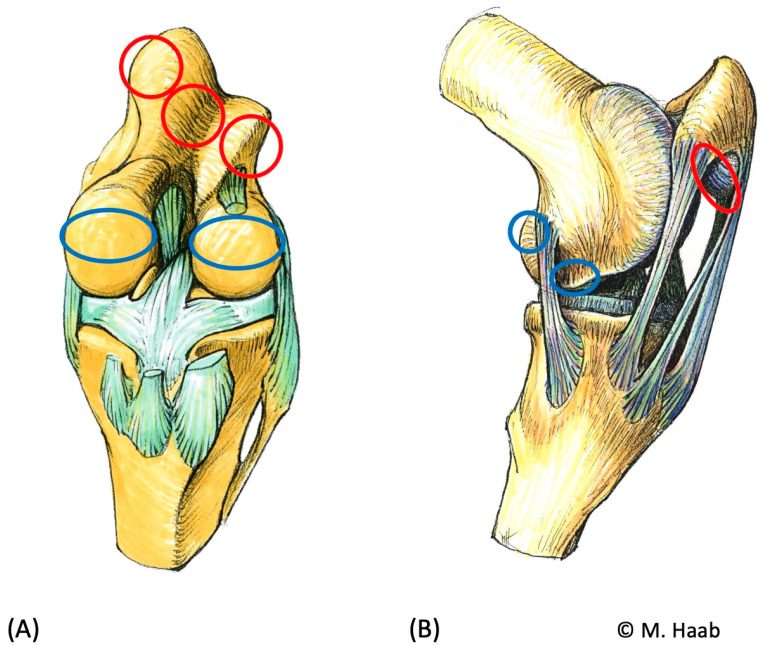
Graphic illustration of a left stifle joint (**A**) cranial view, medial to the left with the patella removed; (**B**) mediolateral view. Red circles correspond to the lesion locations in the femoropatellar joint ((**A**) = trochlear ridges and intertrochlear groove of the femur, (**B**) = facies articularis patellae); blue circles show the lesion locations in the femorotibial joint ((**A**) = cranial femoral condyles, (**B**) = cranial and caudal femoral condyles). On the caudolateral femoral condyle, a comparable lesion location was chosen.

**Figure 2 animals-13-02304-f002:**
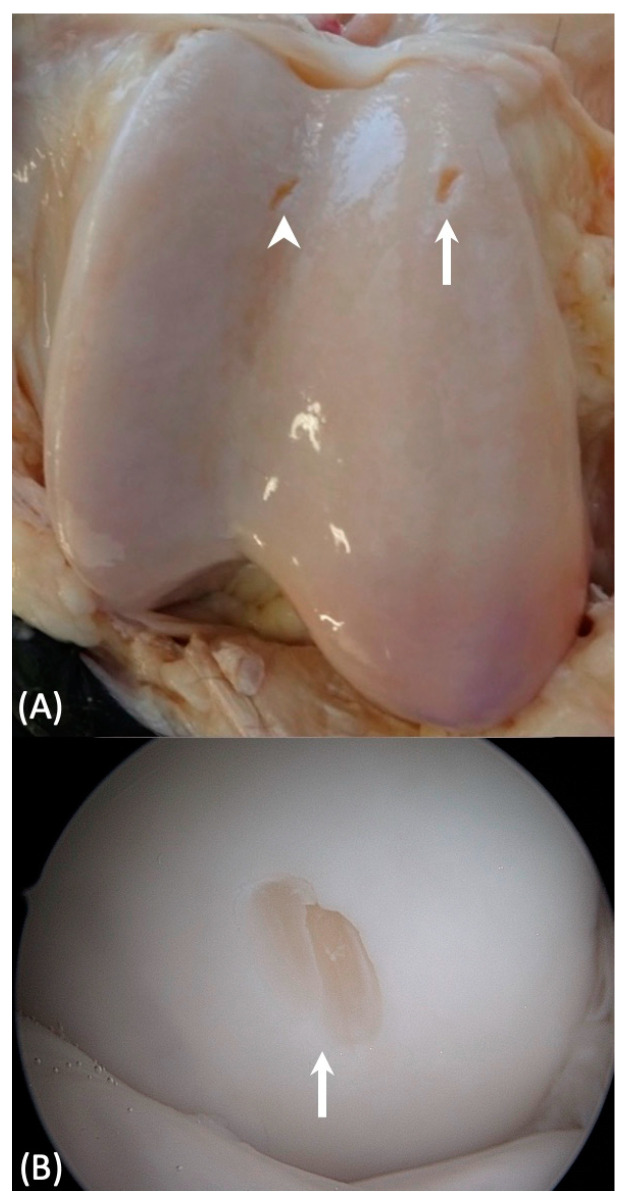
Distal to the top, lateral to the left. (**A**) Macroscopic image of a cartilage lesion on the medial trochlear ridge of the femur (white arrow) and a second lesion in the intertrochlear groove (arrowhead). (**B**) Corresponding arthroscopic view of the medial trochlear ridge lesion (white arrow).

**Figure 3 animals-13-02304-f003:**
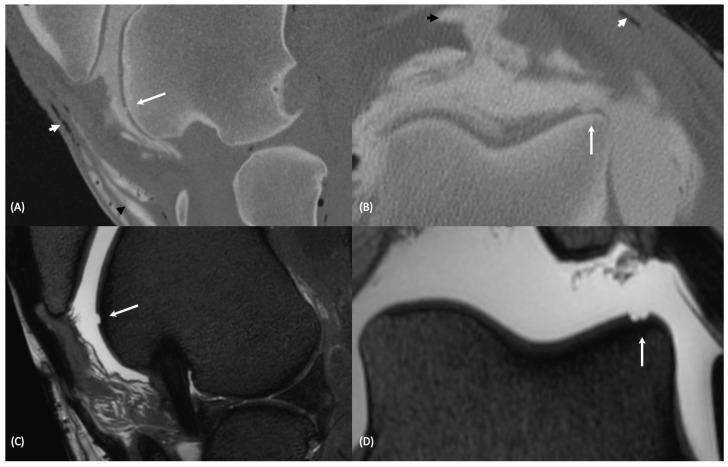
Images showing a cartilage lesion (white arrow) on the lateral trochlear ridge of the femur. CTA images in bone reconstruction algorithm: sagittal plane (**A**), transverse plane (**B**), bottom row 3D T2W MRI images of the same lesion in sagittal (**C**) and transverse (**D**) plane. Subcutaneous gas and contrast accumulation is visible on both CTA images (black arrowhead contrast media, white arrowhead gas artifact).

**Figure 4 animals-13-02304-f004:**
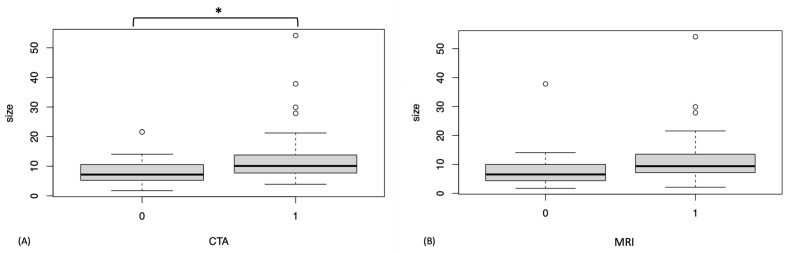
Box plots of mean lesion surface area distribution of arthroscopically created cartilage lesions in equine stifle joints using computed tomographic arthrography (CTA) (**A**) and magnetic resonance imaging (MRI) (**B**). X-axis: 0 = lesion not detected, 1 = lesion detected. Y-axis: mean lesion surface area in mm^2^. Asterisk (*) indicates a statistically significant difference between the mean lesion surface area of detected and not detected lesions.

**Figure 5 animals-13-02304-f005:**
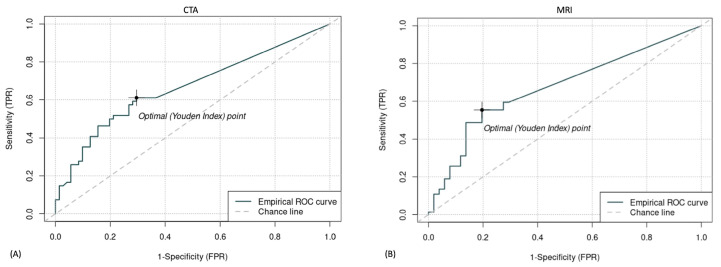
ROC curve with optimal Youden index point using CTA (**A**) and MRI (**B**). The optimal cut-off for lesion detection with CTA was 3.9 mm^2^, resulting in a sensitivity of 61% and a specificity of 70%. The optimal cut-off for lesion detection with MRI was 4.53 mm^2^, resulting in a sensitivity of 55% and a specificity of 80%. TPR = true positive rate. FPR = false positive rate.

**Table 1 animals-13-02304-t001:** Sensitivity and specificity of magnetic resonance imaging (MRI) and computed tomography arthrography (CTA) to detect experimentally created cartilage lesions. n = number of joints included; nc = number of lesions created; nn = number of sites with no lesion created (negative controls); nd = number of correctly detected sites (true positive lesions for sensitivity and true negatives for specificity); CI = 95% confidence intervals in the femoropatellar joint (FPJ), the medial/lateral compartments of the femorotibial joint (FTJ), and in the stifle overall. Significant differences in the McNemar’s tests (*p* < 0.05) are marked with *.

Joint (n/nc/nn)	Sensitivity MRI (nd) CI	Sensitivity CTA (nd) CI	*p*-Value	Specificity MRI (nd) CI	Specificity CTA (nd) CI	*p*-Value
Stifle overall (15/79/68)	66% (52)	53% (42)	0.09	52% (35)	66% (45)	0.04 *
	0.54–0.66	0.42–0.64		0.39–0.64	0.54–0.77	
FPJ (15/35/25)	77% (27)	66% (23)	0.39	36% (9)	56% (14)	0.18
	0.60–0.89	0.49–0.80		0.19–0.54	0.36–0.74	
Medial FTJ (15/27/18)	63% (17)	48% (13)	0.34	61% (11)	72% (13)	0.62
	0.42–0.81	0.29–0.68		0.36–0.83	0.47–0.90	
Lateral FTJ (14/17/25)	47% (8)	35% (6)	0.69	60% (15)	72% (18)	0.45
	0.25–0.71	0.17–0.59		0.4–0.78	0.52–0.87	

## Data Availability

The data presented in this study are available on request from the corresponding author. The data are not publicly available due to privacy issues.

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
