# Peer review of "Diagnostic Performance of Multi-Detector Computed Tomography Arthrography and 3-Tesla Magnetic Resonance Imaging to Diagnose Experimentally Created Articular Cartilage Lesions in Equine Cadaver Stifles"

_animals, 2023, doi:10.3390/ani13142304_

Round 1

Reviewer 1 Report

General comments:

This is an interesting study, with a clear clinical relevance as it supports clinicians to make an informed decision on which imaging modality to choose in case of suspected equine stifle disease. In addition, it allows improved communication with the horse owners regarding the limitations of the different techniques.

In general, I consider this a well-performed study. However, the presentation of the results and the content and flow of the discussion could be improved in my opinion.

I would like to have the authors provide more details regarding the exact locations of the created cartilage lesions and their visibility on the different modalities. Currently only relatively large regions are mentioned eg. medial/lateral femoral trochlea, but not their exact location eg. proximal/mid/distal third, axial vs more abaxial aspects, … similarly for the femoral condyles. Specifically, as this may have had an influence on the visibility in both modalities, this is interesting information for the readers and clinicians as well as for the reproducibility of the study. Therefore, I would like the results described in more detail: how many full thickness and partial thickness lesions per region? How many of each were observed with CTA and how many with MRI. Were the same lesions observed/missed on MRI and CTA or were it different ones? Which were more likely to be observed, which more likely to be missed (eg. abaxial on the femoral condyles). Even if there are no significant differences, there may be a trend. In that case, it would be important, so the clinicians know they have to pay extra attention to specific regions during the evaluation as there is an increased risk to miss lesions in that location. If there was no association with the location of the lesion observed, this is also interesting to mention. Part of this information may be reported in a table. Please also discuss these findings in the discussion section as currently a direct comparison of the performance of CTA and MRI for the same lesion is not possible, which was mentioned as a goal of this study. Please also add if all arthroscopically created lesions were observed on macroscopic evaluation.

The calculation of the lesion’s size and volume is not specified in the materials and methods section. Please clarify this. Specifically, because these lesion are not rectangular in shape. Did these lesion measurements show a normal distribution? If not, the median should be reported apart from the mean value. Please also clarify the calculated cut-off values for lesion size.  The reported values are mentioned in “mm” while all other sizes are mentioned in “mm2”. As the reported values appear small, I think it is unlikely to represent a typing error, but rather represent a unidirectional measurement of the lesion. It would better to use always the same unit, so all size in mm2, including the cut-off for CTA and MRI. Or was this value representing the lesion depth? Please clarify this in the text.

In the discussion, the Outerbridge scale and grading is used to compare with the created lesions in this study to previous studies. However, the way this scale is interpreted and reported in this study, focussing only on the reported size cut off, is misleading in my opinion. Outerbridge grade 1 lesions are lesions characterised by a softening or swelling of the cartilage. Partial thickness (not reaching the subchondral bone) lesions are grade II. Grade III lesions are “fissures” reaching the subchondral bone. As soon as the subchondral bone is exposed, these are grade IV. Therefore, all lesions in this study are at least grade II and all full-thickness lesions are likely grade IV, considering they were created with a curette, so more than a fissure reaching the subchondral bone. Grade IV is only related to exposure of subchondral bone and not to the size of the lesion. Please adjust this in the discussion.

Finally, in the material and methods section, it is mentioned that 1 radiologist evaluated the CTA and MRI images twice with 6 months apart and that the results were compiled. Why were these results compiled and not considered as 2 observations with a intra-observer repeatability calculation? Was there a significant difference between both observations? Please comment on this in the discussion.

Specific comments:

Introduction:

-          Lines 48-50: “It has been … debridement”: this sentence is not well constructed and some words appear to be lacking. Please revise and adjust.

-          Line 90: reference 25 does not support the statement of this sentence.

Material and methods:

-          Line 120: if used as adjectives, the terms proximal and caudal should be used. In accordance to “A STANDARDIZED NOMENCLATURE FOR RADIOGRAPHIC PROJECTIONS USED IN VETERINARY MEDICINE. Veterinary Radiology, Vol. 26, No. 1,1985; pp 2-9.”, these terms can be combined and the sentence adjusted to “… in a slight caudoproximoaxial direction”

-          Line 133: “obturator and cannula were directed….”

-          Lines 145-46: the cranial and caudal aspects of the femoral condyles are mentioned respectively. What is meant exactly? Cranial/caudal half of the condyle? Were the lesions always created at the same level in the cranial aspect? As mentioned in the general comments, the readers would benefit from eg. a drawing showing the distribution and number of the cartilage lesions in the different regions.

-          Line 160: if 350 mg Iodine/ml was diluted 1:4 with saline, the final solution is 70 mgI/ml as 4 units (eg 40 ml) of saline are added to 1 unit (eg. 10 ml) of 350 mgI/mL so the end volume is 5 units (50 ml containing 3500 mgI). However, if 1/4th of the total volume was contrast, the ratio should state 1:3 and the concentration would be 87.5 mgI/ml.

-          Line 174: as far as I know, “ex-articulate” is not a correct term, however disarticulate is commonly used in veterinary and human medicine.

Results:

-          Lines 209, 213: “considered good to excellent by the reviewer”: Which reviewer is meant exactly? Do you mean the radiologist, the primary author, …?

-          Line 210-211: “In two stifles, …only the lateral FTJ was affected”:

o   Please be aware there are only 2 joints in the stifle: the femoropatellar joint and the femorotibial joint, with the latter having 2 compartments. Please adjust accordingly throughout the entire manuscript.

-          Lines 215-217: “Intra-articular air accumulation …” this is a discussion and not a result. The presence of intra-articular gas (number of joints, limbs) was not mentioned as a result/observation in this study before this sentence. The authors also mention that it made the evaluation of the cartilage more difficult, but did it obscure lesions when the images are reviewed again unblinded? Was this only a problem on MRI or also on CTA? Please discuss more in detail (in the discussion section).

-          Line 227: Figure 2 is not representative for this sentence about specificity.

-          Line 228: the joint-specific Se/Sp are shown in table 2, not in table 1. Please adjust.

-          Table 2:

o   I find the current table legend confusing: As it mentions the “Se and Sp of MRI and CTA to detect experimentally created cartilage lesion (n)”, it suggests that (n) are the total number of cartilage lesion created, however it is the number of cartilage lesions detected with this modality for that joint (compartment). The total number of cartilage lesions created are only mentioned in the text and therefore the table cannot be interpreted correctly by itself. Please adjust.

o   In addition, I believe some of the calculations of the percentages may be wrong: stifle overall sensitivity: 51 lesions of the created 79 lesions are seen. This is 64.6% and not 65.8% as mentioned in the table. Same for FPJ 25/35 is 71.4%, not 77.1%. Please recheck all the numbers listed in this table. Specifically, also check if this influences the results regarding the MRI/CTA comparison.

-          Line 229-232: “specificity was good for CTA and moderate for MRI…sensitivity good to mid-range”: What are your references to determine these eg. 66.6% as good and 51.5% as moderate?

-          Line 234: How were the mean lesion size and volume calculated? This is not mentioned in the M&M. Specifically considering these lesions appear dome-shaped and not rectangular or cube shaped.

-          Line 239: “lesions detected by CTA has a significantly greater size: 13.8 mm2 …”. Is this the mean value, or the median or another value? Please add also for the not-detected CTA lesions and MRI.

-          Line 244: Cut-off lesion size for CTA and MRI are mentioned as 3.9 mm and 4.53 mm. What does this value represent? The length or the width, an average of both or depth? Why are these mentioned as mm and not as mm2?

Figures:

-          In Figure 1, line 250, you use the term “trochlear ridge” while in figure 2 line 254 only “trochlea” is used. Please use “trochlear ridge” throughout the entire manuscript for consistency.

-          Figure 2: As the CT images were done after arthrography, also using “CTA” would be more correct in the legend. In addition, this is a bone reconstruction algorithm and not only a bone window. Please adjust. As subcutaneous gas is visible on both images and contrast accumulation also on the sagittal image, it would be informative for non-experienced readers to mention this at least in the legend and possibly indicate it also on the figure.

Table:

-          Table 1:

o   The legend does not mention all the abbreviations used. FH, RL, AP, Sag, Cor, Tra. Please note that some of these planes and directions are from human medicine and not used in veterinary medicine (eg. coronal should be dorsal, AP should be CrCd, …). Please change all these abbreviations to the correct terminology (ref. Smallwood et al. Veterinary Radiology, Vol. 26, No. 1, 1985; pp 2-9 ).

o   Please adjust the width of the columns so the text is interrupted in the correct way: “suppresse    d”; “Sa   g”; “SP    AIR” …

-          Table 2:

o   See above in results section

Discussion:

-          Line 283: “… within equine stifle joint cadavers”: this sounds strange. Please adjust to “within stifle joints of equine cadavers.

-          Line 290, 296: equine carpal joint: the carpus consists of multiple carpal joints, so please use plural.

-          Line 297: the complex equine stifle joint: similarly as for the carpus, the stifle consists of multiple joints. Therefore, it is better to only use the word “stifle” or “stifle joints” in plural.

-          Line 305-306: “lesions created on the femur condyle could get covered by the menisci …”: This is written in a hypothetical way. Was this the cause for the lack of lesion observation in some limbs? It would be interesting for the readers to show also an example of lesions not visualised on CTA and/or MRI. As mentioned above an additional drawing showing more exactly the locations of the created lesions, possibly separated in partial and full thickness, would be of interest for the readers as well as improving the repeatability of the study. Eventually also showing which were seen on MRI and which on CTA would be beneficial for clinical diagnosis of cartilage lesions in patients as is could highlight which are likely to be missed.

-          Line 305: “… places the lesions directly under the meniscus”. As the lesions were created in the femur, which is proximal to the meniscus, the term “under” should be avoided. Please replace by another term.

-          Line 309: “… in different flexion grades …”: please check if this is a correct English expression. I would think “ different degrees of flexion” is more correct, but I am not a native speaker.

-          Line 310:

o   “having a greater density”

o    “… accumulate in the lower-lying joint cavities”: Do you mean “in the more distal located joint cavities” or the “recumbency dependent joint cavities”? Please replace by more specific terminology.

-          Line 312: Why was gravity considered a possible cause for poor sensitivity? As the lateral aspect was the dependent side and the septum between the medial and lateral compartments of the femorotibial joint were interrupted during arthroscopy, I would expect the contrast filling of the medial compartment to be worse than the lateral one. Please clarify.

-          Line 318: While motion will indeed not occur in cadavers, this should be also not be a problem during MRIs of horses under general anaesthesia, specifically not in stifles as traction is usually needed to get the joint within the isocentre of the magnet.

-          Line 322-323: “In artificially … less of CTA”. This sentence nearly completely repeats the content of the line above. Please rephrase this line or consider combining the content of both sentences.

-          Line 324-327: “Whether freezing, gas artifacts and ferromagnetic dust have influenced the diagnostic accuracy … is unclear”. This is a general comment, but did you see any indications that these artefacts caused decreased lesion visibility in this study? The contrast injection was only performed after MRI, so this could not be a cause in this study. Were any lesions observed, considered to be due to ferromagnetic dust from surgical instruments?

-          Lines 348-350: what are considered as severe, moderate and mild cartilage lesions in this reference? Please clarify.

-          Line 368: “If the 3D MRI sequences …, … depend on …”. “if” is used in case of a condition, whether in a choice between multiple options. “Whether” would be preferred in English grammar in this sentence.

-          Line 370: “… and therefore resulting image quality due to motion and signal-to-noise artifacts”. What type of motion are the authors expecting during an MRI scan of a stifle of a live patient in a 3T magnet? Is the signal-to-noise expected to be worse in live patients versus cadaver limbs? Please clarify.

-          Lines 371-373: “… higher contrast concentrations are more easily achieved.”: Were the contrast concentrations achieved in these stifles considered too low and the cause for a lack of lesion detection? Was this the case in specific regions. The used contrast concentration is still double the concentration reported to be useful in other CTA publications, also reported in equine stifle joints. Therefore even a 1:1 dilution of the used contrast concentration  with joint fluid, would still give a more than 40 mgI/mL concentration.

-           Line 377-378: “… affected by this generalised cartilage changes”. Single and plural are combined in this line. Please adjust to “… this … change” or “these … changes”.

-          Lines 378-380: “With dual-energy … equine cartilage”: This comment appears to float and no clear transition from the previous sentence is made. Please revise to improve the flow of the discussion.

Conclusions:

-          Lines 382-383: “CTA … lesion size”. Words appear to be lacking in this sentence. I do not understand the message it is trying to bring. Please rephrase.

-          Line 385-386: While I agree with the content of this line, CTA being used as a non-invasive tool to evaluate the cartilage, I do not think it is correct to state this here in the conclusion. To be able to exclude the presence of a disease, I believe the sensitivity is more important than the specificity. I do not agree that this line is really a conclusion that can be drawn from this study. Please rephrase the conclusion, based on the results from this study.

References:

Multiple names are not correctly listed, some examples below. Please check the correct writing and abbreviations in the original publications.

-          No. 15: one author name is not correctly listed: please replace: van der Veen, H.

-          No. 16: multiple author names are not correctly listed: please replace: Van der Vekens, E. Bergman, E.H.J.; van Bree, H.J.J

-          No. 24: Why is the title placed between brackets?

Mentioned where applicable in the detailed comments.

Reviewer 2 Report

Dear Authors,

I am really happy to have a chance to make the review so interesting topic.

It is a very good direction. CT is a much more simple and faster examination than MRI, so there is a chance to get precise results very quickly.

I know that it is not so easy to perform such a study. All the problems connected with the study have been discussed.

The text needs some small corrections (typos) which I marked in the manuscript. In the section materials and methods its necessary to write precisely which agent gas or fluid was used during the procedure.

Congratulations on doing so a good job.

Reviewer 3 Report

In their manuscript (Diagnostic Performance of Multi-Detector Computed Tomography Arthrography and 3-Tesla Magnetic Resonance Imaging to Diagnose Experimentally Created Articular Cartilage Lesions in Equine Cadaver Stifles), authors imaged cadaver limbs, with CTA and MR, after induction of cartilage defects (full, partial, or no defect). They calculated the sensitivity and specificity for each modality and used a statistical regression model to understand which parameter (joint compartment, lesion size, depth) would influence detection with each modality (CTA, MRI).

As authors mentioned, similar studies has previously been done on other joints (carpus, fetlock), so the design is not novel, but the target (stifle) is. Moreover, the prevalence of such cartilage defect in that joint and the clinical relevance for detecting them make that manuscript very interesting for equine clinician and radiologists.

The main strength of their study is to focus strictly on cartilage defects, as they created cartilage defects, without involvement of the subchondral bone or of other joint tissues. This is very interesting as, in clinic, suspicion of cartilage defect /cartilage abnormalities is sometimes extrapolated from subchondral bone lesions. In that view, the study design really focus on the performances of this diagnosis tool to detect cartilage defects. It is very important since cartilage defect, by them self (without involvement of the subchondral bone) are not painful and, in most case, are progressing and are not self-healing.

The figures are relevant, and I really appreciate the quality of the pictures (macroscopic and imaging).

My main comments are :

(1) One limit of the study: the sample size has not been calculated beforehand for this study about the sensitivity and specificity of a diagnostic tool. This should be acknowledge since there are is some analysis ‘close to significance’ findings (P=0.08, could change into P=0.04 or P=0.12 if increasing sample size).

It is likely that the authors could, a posteriori, calculate an observed power for their study (for example: using a programm like G*Power that enable both : calculate sample size (before starting a study), but also calculate the power of a study (when sample size is known, and outcomes are available), the latter would increase the description of the results strength by showing the size of the alpha error (p-value, provided in the manuscript) and the size of the beta error (1-power; not provided in the manuscript).

(2) In Table 2, I am not sure to get it right: Is the 95% confidence interval calculated based on the 2 analysis (6-months apart) by the radiologists? In case there are only 2 values to compute this CI, I would go for mentioning the two values, without calculation of a mean/CI.

Moreover, in the first column, should the n indicated in brackets as (51) be 50 ? I thought it was 25 (FPJ) + 17 (medial FTJ) + 8 (Lateral FTJ) = 25+17+8=50. In case I am wrong, it would worth an explanation in the footnote in order to avoid other readers to do the same misinterpretation.

 (3) In material and methods: There seemed to be 2 assessments by the same radiologist, did the authors calculate a kappa value for intra-observer agreement? It is interesting to see how the two observations would be consistent or not, using those techniques, with an expert in radiology.

Thank you for this nice manuscript.

Reviewer 4 Report

In both short summary and Asbtract, the maths are confusing, why not immediately enumerate the total number of legs (19) collected and included (as free of pre-existing cartilage lesions) from the total number of horses (15)?

Line 142: it is contradictory to say that the defect size is randomly determined by the surgeon but at the same time depended the accessibility and the angle of the curette

Table 1: is it relevant to have it there? Can it be simply added as supplementary

Line 194: you state "Arthroscopy served as the reference". Can you elaborate reference for what? Presence or absence of lesions? It is not mentioned earlier that sites and dimensions of lesions created during arthroscopy were recorded. Was an objective and exact measurement of the lesions created possible during arthroscopy procedure?

Line 206: this contradict line 15 of the summary where it is mentioned 15 horses, while the actual total horses included is 11. please review the maths to be coherent at the different sections

Paragraph starting line 218 suggests that an unequal number of lesions were created per selected joint. Is there a justification for this?

Line 234 and full paragraph "size" is possibly too vague and be replaced by "surface" or "surface area" that is more appropriate as being more specific 

Line 328. see above "size" swapped with "surface" or "surface area"

Line 365: This is probably one of the major limitations of the project Only one specialist radiologist (although blinded) was involved in the subjective assessment of the images. The same specialist performed the similar image analyse twice at 6 months interval. Wouldn't it have been preferable to involve 2 or 3 different DI specialists? How blinded and randomised was the second assessment relative to the first?

Line 372: contrast concentration is defined by the dilution practitioners are deciding, how is this different between research and practice?

Lines 15 and 29: the construction of sentence is a bit unusual. would it be preferable to start with "A total of 79 cartilage defects were created arthroscopically in...."

Line 59: sentence construction is grammatically incorrect with inversions that are not usual for English language "early detection of cartilage defects..."

Line 111: swap thawed with defrosted

Line 133: typo "war" to be corrected to "was"

Line 63 : add a comma after "in horses"

Line 237: authors probably meant "neither" rather than "whether" at the start of this sentence

Line 303: sentence construction makes is rocky to read

Line 322: sentence could be reviewed for a more fluid

Round 2

Reviewer 1 Report

Please see changes in uploaded review-2. All new comments are added in blue.

Minor editing and review by a native English speaker would be beneficial.

Round 3

Reviewer 1 Report

Mentioned in the review.

Author Response

Dear Reviewer

Thank you for your review. We adapted the manuscript as requested. 

Lesion location is now used to describe the 8 different areas of lesion creation.

The legend has been corrected with n= number of joints

All minor comments were changed as requested.

A native English speaker has made some minor adjustments to the sentences: 

Line 12: "In order to prevent the onset of osteoarthritis"

  Line 54: "The timely detection of cartilage damage in the joint disease process is essential, to allow prompt treatment, restore joint functionality, and prevent, or delay the onset of osteoarthritis (OA) [1,2]."